

# Dense symmetric temporal alignment learning for human pose estimation

Guang Xv[1] and Xingchen Wu[2]

[1] Shenyang Aerospace University, Shenyang, China
[2] Jiushao Institute of AI Algorithm, Jihua Laboratory, Foshan, Guicheng, China

## ABSTRACT

Human pose estimation aims to locate the human joint positions from images or videos. This problem has drawn increasing attention and wide applications in autonomous driving, motion analysis, and intelligence robotics. Some existing works aggregate movement features from neighbouring frames, which is instrumental in capturing sufficient information. However, considering the fast motion and pose occlusion in videos, directly incorporating unaligned additional visual cues from adjacent frames is prone to introduce noises due to the significant differences in inter-frame characteristics. In this article, we advocate executing adequate feature alignment between the keyframe and supporting frames to better utilize neighboring frame contexts. Towards this end, we propose a novel symmetric U-Net-like feature alignment algorithm for the human pose estimation task. This algorithm learns symmetric information at global and local levels for each scale separately to assist the model in generating accurate results. Specifically, a global alignment block based on temporal deformable convolution is designed to learn the complex temporal dynamics between adjacent and current frames to align the features. Moreover, a local alignment block based on adaptive convolution is presented to optimize the feature information further and preserve the geometry structures. Coupling these two modules into a U-Net-like symmetric architecture forms our framework. We show the effectiveness of our algorithm through the excellent results on two large pose estimation benchmark datasets: PoseTrack2017 and PoseTrack2018. In addition, we demonstrate that the proposed model achieves state-of-the-art performance on the self-built badminton dataset.

## INTRODUCTION

Human pose estimation, which has attracted increasing attention in the past decade, is to locate the human configuration from input data, including images and videos. The ability of machines to estimate and find human joints assists them in understanding the behavior of human beings, which is significantly important to interact safely and reasonably with people. As a result, it is hugely coveted in a broad spectrum of applications, such as human-computer interaction, autonomous driving, intelligence robotics, and animation production (*Basurto et al., 2024*; *Fang et al., 2023*; *Rohini et al., 2025*).

Recently, approaches explored employing various deep learning-based methods to solve this task, such as convolutional neural networks (CNN), generative adversarial networks

Corresponding author
Xingchen Wu,
xingchenwu0104@gmail.com

(GAN), and Transformer. Existing models can be divided into image-based and video-based human pose estimation. The early attempts focus on tackling image-based pose estimation tasks (*Wei et al., 2016*; *Fang et al., 2017*). Numerous image-based pose estimation models have achieved positive performance, especially in computational efficiency. However, directly utilizing image-based algorithms to tackle video-level tasks frequently generates unsatisfactory results due to the failure to extract the temporal-dependent relationship in the video. Specifically, when occlusion or motion blur occurs, these networks have difficulty capturing complementary information from adjacent frames, which brings negative impacts on achieving accurate pose estimation (*Feng et al., 2022*). Consequently, the video-based human pose estimation models have been designed to encode adequate motion information from video sequences *via* various temporal feature processing modules for improving pose estimation performance. Existing video-based algorithms directly aggregate information from neighboring frames to enhance the feature representations of the target frame (*Fang et al., 2023*; *Peng, Zheng & Chen, 2024*; *Mehraban, Adeli & Taati, 2024*; *Naseer et al., 2024*; *Kumar & Naganaik, 2023*). However, the rapid movement of the human body causes significant differences between the information at the same position in the adjacent frames. So, the direct aggregation information method inevitably introduces negative features, such background noise and irrelevant characters, which brings challenges for efficiently capturing human body features from neighboring frames (*Liu, Li & Huang, 2024*; *Liu et al., 2023*; *Naseer et al., 2023*; *Naseer, Khan & Altalbe, 2023*; *Khan et al., 2024*).

To solve the shortcomings of existing algorithms, we proposed to capture correlative human information from consecutive frames to enhance the pose estimation performance in videos. Our method, termed dense symmetric temporal alignment learning method for human pose estimation (DSA), which contains a series of parallel global information alignment blocks and a local information enhancement blocks. In contrast to existing methods that directly perform multi-frame aggregation, we propose to densely align the features of supporting frame to the key frame, there by improving the usage of temporal information and facilitating the model's robustness in challenging cases such as occlusion or blur. Moreover, we design a global-to-local alignment paradigm to process large motions and local details, respectively. We also combine the proposed alignment method with a symmetric learning framework, enabling capturing more enriched representations.

The contributions of our method are summarized as follows:

(1) We design a global information alignment block to explore global related connections between the current and adjacent frames to coarse align them, which can aggregate more satisfactory information.

(2) We propose a local information enhancement block to encode local detail information across frames for fine-grained feature alignment, which able to effectively deliver comprehensive representation.

(3) Extensive experiment results show that our method achieves excellent performance on three human pose estimation datasets, including PoseTrack2017, PoseTrack2018, and the self-constructed badminton dataset.

## RELATED WORK

### Image-based human pose estimation

Human pose estimation has gained significant attention in recent years. For example, *Wei et al. (2016)* proposed a sequential structure model based on CNN, which generated more accurate results through iterative optimization of each module in the training process. *Fang et al. (2017)* designed a symmetric spatial feature encoding network to capture high-quality pose information from inaccurate bounding boxes. *Sun et al. (2019)* built a high-to-low resolution model to maintain multi-resolution mapping features through feature extraction, which fuses various resolution representations to improve estimation accuracy. *Moon, Chang & Lee (2019)* proposed a fully learning-based camera distance-aware top-down general framework compatible with most existing human body detection and estimation modules. *Kreiss, Bertoni & Alahi (2019)* introduced a bottom-up method for multi-person 2D pose estimation, which leveraging part intensity fields (PIF) for body part localization and part association fields (PAF) for pose formation. *Maji et al. (2022)* developed You Only Look Once (YOLO)-pose, a heatmap-free method for 2D pose estimation, optimizing the object keypoint similarity metric end-to-end and detecting poses in a single forward pass.

### Video-based human pose estimation

Nevertheless, utilizing image-based approaches to video may leads to reduced performance, due to motion blur, occlusion, and similar challenges. For solving these problems, *Song et al. (2017)* extracted relevant movement features from the temporal domain by calculating the dense optical flow features between adjacent frames. *Luo et al. (2018)* designed an end-to-end architecture that employed a loopy spatio-temporal graph to encode the consistency of human poses in videos. *Bertasius et al. (2019)* proposed a pose warper network based on a space-time distortion mechanism, which improves the label propagation between frames and benefits the training from sparsely labeled videos. *Fang et al. (2023)* combined symmetric integral keypoint regression and pose aware identity embedding, enhanced by PGPG and knowledge distillation, to accurately localize human keypoints. *Peng, Zheng & Chen (2024)* introduced a dual-augmentor framework that enhances pose estimation generalization through meta-optimization and differential strategies. *Mehraban, Adeli & Taati (2024)* built on AGFormer blocks that combine transformer and GCNFormer streams to enhance 3D structure learning through adaptive fusion, offering four variants for speed-accuracy trade-offs. *Li et al. (2025)* designed a unified MLP-GCN architecture for 3D pose estimation, which efficiently modeling spatial and temporal dynamics with minimal computational cost. *Li et al. (2024)* proposed hourglass tokenizer (HoT) for efficient 3D pose estimation, including a token pruning cluster to reduce redundancy and a token recovering attention to restore full-length temporal resolution.

## OUR APPROACH

**Problem formulation**. Presented with a current frame $I_t$ and its neighboring frame $I_n = \langle I_{t-2}, I_{t-1}, I_{t-1}, I_{t+2} \rangle$, we aim to estimate the human joint position of frame $I_t$. Our

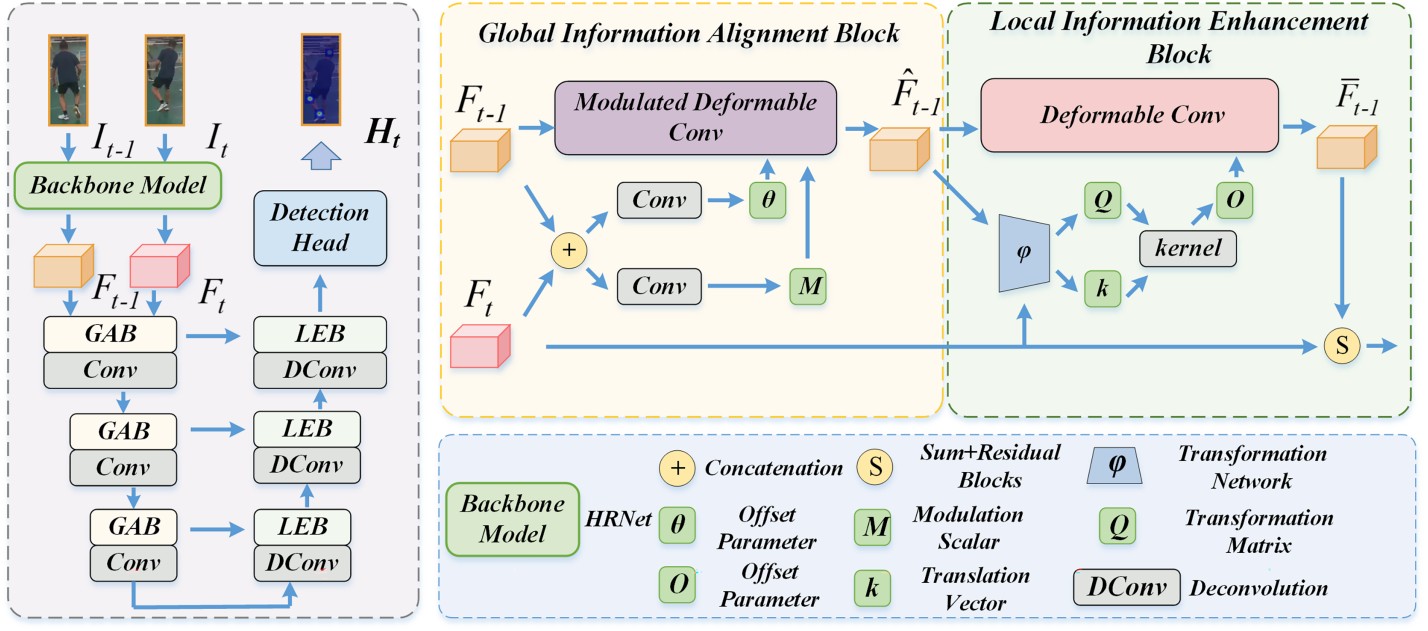

**Figure 1  Overall pipeline of our DSA framework.**

goal is to better leverage temporal information from the neighboring frame through the proposed global and local feature alignment, thereby addressing the common shortcomings of existing approaches in failing to effectively utilize closely related information.

**Method overview**. The pipeline of our designed dense symmetric temporal alignment (DSA) learning method is illustrated in Fig. 1. The proposed method belongs to the top-down framework that first uses a person detector to expose each individual in the video and then performs the task to a simpler single-person pose estimation problem. Specifically, we first employ the Faster-RCNN (*Pankajakshan & Bhavsar, 2020*) as a person detector to obtain the bounding box of the human pose at current frames. The bounding box is enlarged 25% to ensure that the model captures the pose features at the same position from the neighboring frames. Then, the pose features are feed to our DSA model for symmetric aggregating both globally aligned information and local fine-grained visual context to generate more comprehensive representations. Finally, a detector head is employed to output the heatmap result of the current frame. Existing methods overlook the misaligned spatial contexts of different frame, leading to suboptimal performance. We propose a global to local alignment method, and design global information alignment block which aligns the global information and focuses large motion contexts. The local information enhancement block further processes the unaligned local details while retaining the geometric structure of human keypoints.

## Global information alignment block
Ideally, incorporating features from similar positions of adjacent frames will obtain the optimal representation of the human pose to improve estimation accuracy. However, the

rapid movements of persons or cameras cause significant spatial shifts of person between each frame. Existing algorithms direct fuse features from neighboring frames, which inevitably introduce noise information that is unrelated with the current frame. We instead propose the global information alignment block (GAB) to obtain a coarse preliminary alignment of the supporting frame. The GAB computes the spatial alignment parameters of the global affine transform to achieve coarse preliminary alignments between neighboring frames. It is worth noting that, unlike existing methods that perform feature alignment using several deformable convolution layers, we employ a symmetric architecture that repeatedly employs deformable convolutions to achieve more adequate feature alignment.

As we know, the simple linear mapping or regular convolution network can be empoyled to capture spatial features, but it fails to extract satisfactory information from the complex and variable human poses. In contrast, the deformable convolution network able to adaptively transform the sampling point of the convolution kernel to selectively encode information from various spatial locations. Motivated by this, we design GAB based on deformable convolution network to globally align the supporting frames $I_s = \langle I_{t-2}, I_{t-1}, I_{t-1}, I_{t+2} \rangle$ with the current frame $I_t$ at coarse level.

To facilitate the representation of this operation process, we take the alignment stage of neighboring frame $I_{t-1}$ as an example.

We first utilize the high-resolution network as the backbone, the state-of-the-art image-based human pose estimation method, to extract the features from the current frame and the neighboring frames to obtain the feature vector $F_t$ and $F_{t-1}$, respectively. The HRNet is pre-trained on the human pose estimation dataset (COCO dataset) to obtain high-resolution human features. The feature vector $F_t$ and $F_{t-1}$ feed our GAM to compute global rearrangement parameters to obtain a preliminary alignment of the supporting frame feature with the current frame feature.

Specifically, the process stage of GAB is as follows:

Given $F_t$ and $F_{t-1}$, the sampling offsets of each pixel in the convolutional kernel $\theta$ and the modulated scalars that denote the sampling magnitude $M$ of global affine transform phase can be obtained as follows:

$$\begin{aligned} \theta &= \mathbb{T}_\theta \left( F_t \oplus F_{t-1} \right) \\ M &= \mathbb{T}_M \left( F_t \oplus F_{t-1} \right) \end{aligned} \tag{1}$$

where $\mathbb{T}_\theta$ and $\mathbb{T}_M$ denote the transform nodel that generates $\theta$ and $M$, respectively. These models have the same structure consist of two basic modules with $3 \times 3$ convolution kernel, which are trained independently without sharing parameters. $\oplus$ denote the concatenation operation of the features by channel dimension. $\theta = \langle \Delta p_n | n = 1, 2, 3, \ldots, |\Re|. \rangle$ denotes the offset value of each pixel position on the convolution kernel, $\Re$ denotes a standard $3 \times 3$ convolution kernel, which contains nine pixels with the coordinates of $\Re = \langle (-1-1)(-1,0)(-1,1) \ldots (0,1)(1,1) \rangle$.

Subsequently, we input the spatial offset parameter $\theta$, the modulation scalar $M$, and the neighboring frame features $F_{t-1}$ to the deformable convolution model $f(\bullet)$ for
generating the globally aligned coarse information $\hat{F}_{t-1}$, which can be formulated as follows:

$$\hat{F}_{t-1} = f(F_{t-1}, \theta, M). \tag{2}$$

For a pixel $p_0$ in the coarse aligned feature vector $\hat{F}_{t-1}$ at the rearrangement transformation stage can be formulated as:

$$\hat{F}_{t-1}(p_0) = \sum_{p_n \in \Re} w(p_n) F_{t-1}(p_0 + p_n + \Delta p_n) M \tag{3}$$

where $w(p_n)$ denotes the weight of pixel $p_n$ in the convolution kernel, and $p_n + \Delta p_n$ indicates the new sampling position of convolution kernel after deformation process modulating.

## Local information enhancement block

The global information alignment block achieves coarse calibration to preliminarily rectify spatial shifts or jitter. However, GAB fails to yield satisfactory alignment for some small parts (*e.g.*, hands and feet), which causes this part feature captured from the supporting frame $\hat{F}_{t-1}$ to be weakly associated with the current frame $F_t$. Moreover, the model keeps the original geometric structure of the feature during information extraction which is beneficial to improve the robustness of the feature representation and enhance the pose estimation accuracy. So, We further propose a local information enhancement block (LEB) to align local details and produce meticulous fine-tuning at pixel level.

Specifically, taken $\hat{F}_{t-1}$ and $F_t$ as inputs, we employ an extended spatial transformation network $\varphi$ to computer a transformation matrix $Q \in \mathbb{R}^{2 \times 2}$ and a translation vector $k \in \mathbb{R}^{2 \times 1}$ from the position parametric of pixels.

$$Q, k = \varphi(\hat{F}_{t-1}, F_t). \tag{4}$$

During this process, the current frame feature matrix $F_t$ is only used as reference features to participate in the feature propagation process, and their feature values are not changed.

Then, the adaptively trained transformation matrix $Q$ and translation vector $k$ utilized to obtain a set of kernel offset parameters $O = \langle o_1, \ldots, o_9 \rangle$.

$$O = Q\Re + k \tag{5}$$

where $\Re$ denotes the $3 \times 3$ kernel of conventional convolution.

Subsequently, we achieve the local rearrangement operation of each pixel through the adaptive convolution network to obtain enhanced feature $\bar{F}_{t-1}$. Specifically, given the coarse calibrated feature $\hat{F}_{t-1}$ and the offset parameters $O$, the computation of each pixel $q$ in $\bar{F}_{t-1}$ can be expressed as follows:

$$\bar{F}_{t-1}(q) = \sum_{o_i^q \in O} w_i \hat{F}_{t-1}(q + o_i^q). \tag{6}$$

For the pixel $q$ in the output features $\bar{F}_{t-1}$, the original convolution aggregates the $3 \times 3$ pixels (in the kernel location) around $q$ in input features $\hat{F}_{t-1}$. In contrast, for each pixel

location in the original convolutional kernel, an offset $o$ is added to adjust the sampling (convolution) position in the input features $\hat{F}_{t-1}$.

We would like to point out that employing the affine transformation to rearrange information of adjacent frames able to improve the robustness of feature representation and preserve the geometric structure of local details.

## Heatmap generation

Finally, we concat all global-local aligned neighboring frame features and the current frame feature at the channel dimension and then employ several residual blocks to aggregate them together to generate improved comprehensive information. We feed them to a detection head to output the heatmaps of each person at the current frame. Note that we utilize a stack of $3 \times 3$ convolution layers as the detection head. By effectively capturing closing related information from supporting frames *via* our dense symmetric temporal alignment learning method, our DSA is more suitable for solving visual degeneration tasks and generating more accurate pose estimation results.

## Loss function

The standard pose estimation loss function (*Jiao et al., 2022*; *Liu et al., 2022*) is adopted to supervise the learning of final pose estimates. The goal is to reduce the total Euclidean or L2 distance between the estimation and the ground truth heatmaps. The loss function can be formulated as follows:

$$L = \frac{1}{J}\sum_{j}^{J} v_j \left\| H_j - G_j \right\| \tag{7}$$

where $H_j$ and $G_j$ denote the estimated and ground truth heatmap of joint $j$, respectively. $v_j \in (0, 1)$ is visibility of joint $j$. The total number of joints in each person is $J = 15$.

# EXPERIMENTS

In this section, we evaluate our algorithm on three pose estimation benchmark datasets, including PoseTrack2017, PoseTrack2018, and the self-constructed badminton dataset. We first present the experimental settings, including datasets, evaluation metrics, and implementation details. We then compare our DSA with existing pose estimation methods in terms of quantitative results and visual results. Finally, we introduce ablation studies to examine the effectiveness of our proposed component in our algorithm.

## Experimental settings

### Datasets

**PoseTrack dataset** is the largest public benchmark dataset for video-based human pose estimation tasks. The PoseTrack2017 consists of 514 videos with a total of 16,219 frames of human pose annotations, which are split into 250 video sequences for training and 50 video sequences for validating (*Yang et al., 2021*). The PoseTrack2018 includes 1,138 videos with 153,615 pose annotations. These are divided into 593 sequences for training and 170 sequences for validation, respectively (*Sun et al., 2019*).

**Badminton dataset**. We utilize a monocular camera to record multi-player badminton match videos. Then, HRNet (*Sun et al., 2019*) has been employed to preliminary localize the human joints. Finally, the preliminary annotated data is refined by professional data annotation workers to annotate the human body posture, following a unified metric of 15 keypoints. For each labeled image, three inspectors will cross-check to ensure the accuracy of the labeling. Unsatisfactory annotations will be redone. The badminton dataset contains a total of five long video sequences with 13,000 frames. These are split into 10,000 and 3,000 videos respectively for training and testing.

Each pose of datasets contains 15 annotated joints and an additional visibility label for each joint.

### Evaluation metric

We evaluate our proposed method *via* the standard pose estimation metric (*Cuiping, 2021*; *Kulkarni & Shenoy, 2021*; *Sun et al., 2019*; *Bao et al., 2020*), termed the average precision (AP). We first calculate the average precision of each joint, and the average over all joints is obtained as the final performance (mAP). Note that we merely use the visible joints to compute the performance value.

### Implementation details

Our DSA network is implemented on PyTorch and experimented on 2 Nvidia Tesla P40 GPUs. For training, we employ some data augmentation strategies, including random rotations $[-45°, 45°]$, random scale $[0.65, 1.35]$, truncation, and horizontal flip. Input frame size is $384 \times 288$. The Adam Optimizer is employed with a base learning rate of $1e-3$ which decays by 10% every 10 epochs. Batch size is set to 48 and trained for 20 epochs. The extract window of supporting frames is set to $(-1, -1, 1, 2)$. We utilize the HRNet-W48 (*Sun et al., 2019*) model to extract visual feature, which is pre-trained on the COCO dataset.

## Comparison with existing pose estimation methods

### PoseTrack2017

We first evaluate our model on PoseTrack2017 validation set with the widely adopted AP metric. Table 1 shows the quantitative results of the above methods in terms of AP and mAP on the PoseTrack2017 validation set. A total of eight methods are compared, including PoseTracker (*Girdhar et al., 2018*), PoseFlow (*Xiu et al., 2018*), JointFlow (*Doering, Iqbal & Gall, 2018*), FastPose (*Zhang et al., 2019*), SimpleBaseline (ResNet-50) (*Zhang et al., 2020*), STEmbedding (*Jin et al., 2019*), HRNet (*Sun et al., 2019*), and our DSA. The proposed DSA framework consistently outperforms these approaches and achieves a mAP of 77.8. Significantly, our model obtains encouraging performance for the challenging joints: we achieve a mAP of 73.6 for the wrist and an mAP of 70.0 for the ankle. Such consistent and significant improvements show that incorporating meaningful characters from supporting frames outperforms methods that use a single current frame.

**Table 1** Performance comparisons on the PoseTrack 2017 validation set.

| Method | Head | Shoulder | Elbow | Wrist | Hip | Knee | Ankle | mAP |
|---|---|---|---|---|---|---|---|---|
| PoseTracker (*Girdhar et al., 2018*) | 67.5 | 70.2 | 62 | 51.7 | 60.7 | 58.7 | 49.8 | 60.1 |
| PoseFlow (*Xiu et al., 2018*) | 66.7 | 73.3 | 68.3 | 61.1 | 67.5 | 67 | 61.3 | 66.5 |
| JointFlow (*Doering, Iqbal & Gall, 2018*) | – | – | – | – | – | – | – | 69.3 |
| FastPose (*Zhang et al., 2019*) | 80 | 80.3 | 69.5 | 59.1 | 71.4 | 67.5 | 59.4 | 69.6 |
| ResNet-50 (*Zhang et al., 2020*) | 81.7 | 83.4 | 80 | 72.4 | 75.3 | 74.8 | 67.1 | 76.4 |
| STEmbedding (*Jin et al., 2019*) | 83.8 | 81.6 | 77.1 | 70 | 77.4 | 74.5 | 70.8 | 76.5 |
| HRNet (*Sun et al., 2019*) | 82.1 | 83.6 | 80.4 | 73.3 | 75.5 | 75.3 | 68.5 | 76.9 |
| AlphaPose (*Fang et al., 2023*) | – | – | – | – | – | – | – | 76.9 |
| DSA | 82.9 | 84.2 | 80.9 | 73.6 | 76.8 | 76.2 | 70.0 | 77.8 |

**Table 2** Performance comparisons on the PoseTrack 2018 validation set.

| Method | Head | Shoulder | Elbow | Wrist | Hip | Knee | Ankle | mAP |
|---|---|---|---|---|---|---|---|---|
| STAF (*Raaj et al., 2019*) | – | – | – | 64.7 | – | – | 62.0 | 70.4 |
| AlphaPose (*Fang et al., 2017*) | 63.9 | 78.7 | 77.4 | 71.0 | 73.7 | 73.0 | 69.7 | 72.5 |
| TML++ (*Hwang et al., 2019*) | – | – | – | – | – | – | – | 74.6 |
| MDPN (*Guo et al., 2018*) | 75.4 | 81.2 | 79.0 | 74.1 | 72.4 | 73.0 | 69.9 | 75.0 |
| PGPT (*Bao et al., 2020*) | – | – | – | 72.3 | – | – | 72.2 | 76.8 |
| Dynamic-GNN (*Yang et al., 2021*) | 80.6 | 84.5 | 80.6 | 74.4 | 75.0 | 76.7 | 71.8 | 77.6 |
| AlphaPose (*Fang et al., 2023*) | – | – | – | – | – | – | – | 74.7 |
| DSA | 81.5 | 85.1 | 81.4 | 74.8 | 75.6 | 77.1 | 72.3 | 78.3 |

### PoseTrack2018

We also benchmark our model with other methods on the PoseTrack18 dataset. Quantitive results on the validation set are tabulated in Table 2, including STAF (*Raaj et al., 2019*), AlphaPose (*Fang et al., 2017*), TML++ (*Hwang et al., 2019*), MDPN (*Guo et al., 2018*), PGPT (*Bao et al., 2020*), Dynamic-GNN (*Yang et al., 2021*), and DSA. As shown in this table, our approach once again delivers the best performance. Specifically, we can observe that our model shows a remarkable 78.3 mAP on the validation set, achieving a 0.7 mAP gain over the Dynamic-GNN (*Yang et al., 2021*). The estimation results of relatively difficult joints, such as elbow, wrist, and ankle, consistently outperformed other methods. These results demonstrate that our model effectively extracts complementary information from adjacent frames and achieves fine-grained feature alignment, significantly enhancing estimation accuracy.

Furthermore, we display the visual results for complex scenarios of our algorithm in Fig. 2, which demonstrate the importance of our method in embracing complementary cues from neighboring frames. As shown in Fig. 3, our method achieves more robust results in challenging cases such as occlusion and blur.

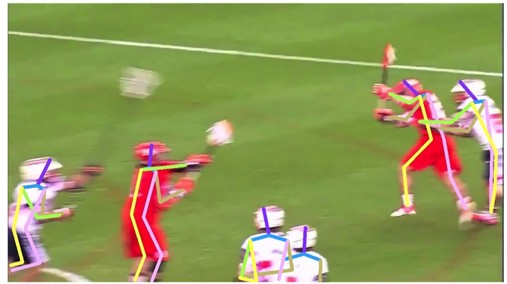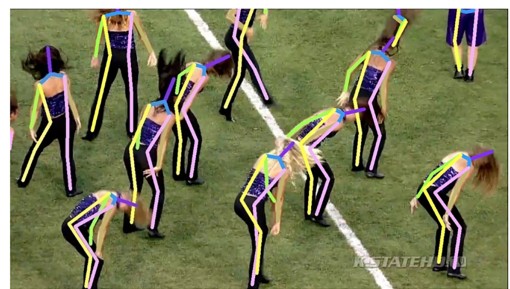
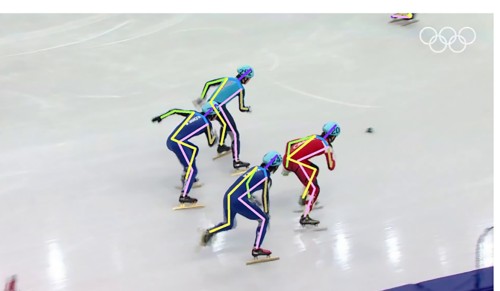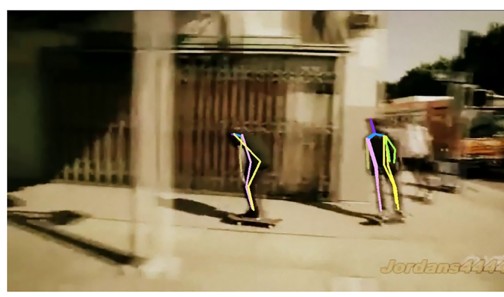

**Figure 2  Visual results of our algorithm on the PoseTrack 2017 and 2018 dataset.**

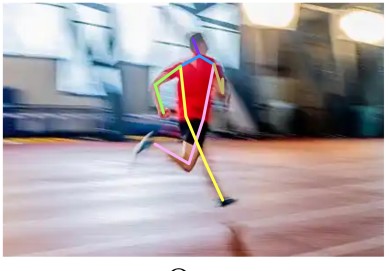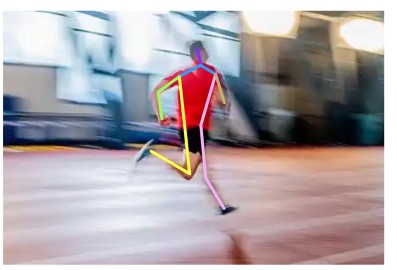

Our                                        HRNet

**Figure 3  Visual results of our algorithm on challenging case of PoseTrack 2017 and 2018 dataset.**

### Badminton dataset

To further evaluate the proposed method, we compare DSA with existing methods in our self-constructed badminton dataset, including CPM (*Wei et al., 2016*), LSTM-PM (*Luo et al., 2018*), ResNet-50 (*Zhang et al., 2020*), HRNet (*Sun et al., 2019*), and our DSA. The evaluation results are tabulated in Table 3. We observe that HRNet (*Sun et al., 2019*) has achieved an impressive accuracy of 70.3 mAP, while our proposed method obtains the best performance of 72.8 mAP on this dataset. We also achieve a 79.8 and 78.5 mAP for the Head and the Shoulder joint, respectively. As illustrated in Fig. 4, we display the visual results on the badminton dataset.

### Ablation study

We perform ablation experiments focused on investigating the contribution of each component in the proposed DSA algorithm, including a global information alignment

**Table 3 Performance comparisons on the badminton dataset.**

| Method | Head | Shoulder | Elbow | Wrist | Hip | Knee | Ankle | mAP |
|---|---|---|---|---|---|---|---|---|
| CPM (*Wei et al., 2016*) | 70.1 | 61.5 | 69.0 | 66.5 | 55.6 | 59.8 | 51.2 | 61.9 |
| LSTM-PM (*Luo et al., 2018*) | 71.2 | 63.5 | 69.2 | 67.0 | 57.0 | 61.2 | 52.0 | 63.0 |
| ResNet-50 (*Zhang et al., 2020*) | 72.3 | 71.0 | 70.5 | 69.8 | 62.1 | 66.4 | 57.8 | 67.1 |
| HRNet (*Sun et al., 2019*) | 76.8 | 73.2 | 71.1 | 70.5 | 68.6 | 69.8 | 59.8 | 70.3 |
| DSA | 79.8 | 78.5 | 75.6 | 71.2 | 70.8 | 72.3 | 61.9 | 72.8 |

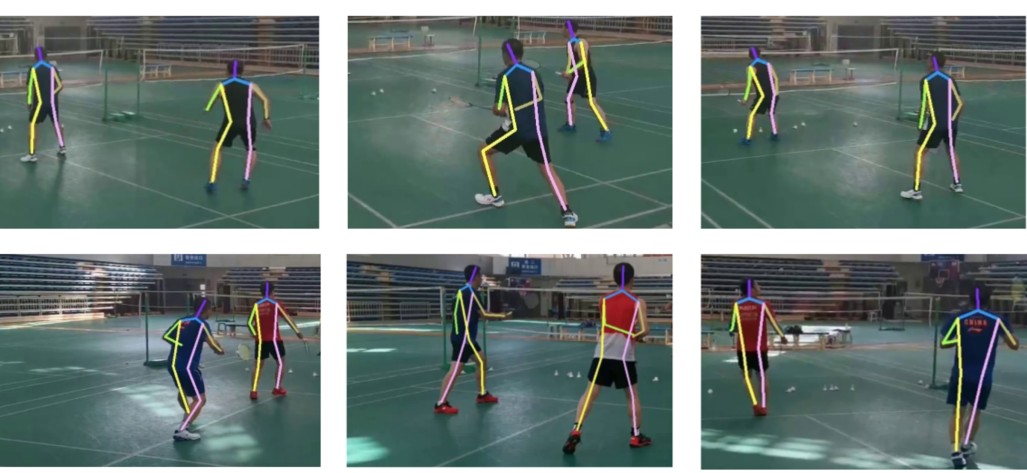

**Figure 4 Visual results of our algorithm on the badminton dataset.**

block (GAB), a local information enhancement block (LEB), and U-Net-like architecture. All experiments are conducted on the PoseTrack2017 validation set. All experiments are performed on the PoseTrack2017 validation set, with results reported in Table 4.

### Global information alignment block

We investigate the influences of employing different number of supporting frames for feature aggregation, where the number is set to 2, 6, and 4 (complete model). As presented in the first and second lines of Table 4, we observe that an unreasonable number of supported frames causes negative impacts to pose estimation. The model utilized two supporting frames with 75.9 mAP, this reveals that the estimation accuracy is reduced due to the insufficient supporting information contained in the shorter sequence. The model employed six supporting frames with mAP drops of 0.3, which suggests that aggregate information from long-distance frames introduces irrelevant information. As shown in the third and sixth lines, the algorithm fails to adapt to feature geometry in the coarse alignment stage causing a slight degradation in estimation accuracy. The GAB has been removed the mAP falls from 77.8 to 74.3. This significant performance drop demonstrates the important role of this component in the feature extraction process. We conduct an ablation study on the temporal alignment. From the results in rows 6 and 7 of Table 4, the temporal alignment brings significant performance improvements.

**Table 4 Ablation studies of different components in DSA performed on PoseTrack2017 validation set.** "r/m X" denotes removing X module in the framework.

| Method | mAP |
| --- | --- |
| Two neighboring frames | 75.9 |
| Six neighboring frames | 77.5 |
| GAB with regular convolution layer | 77.2 |
| r/m HAM | 74.3 |
| r/m LEM | 76.5 |
| r/m alignment | 72.6 |
| DSA, complete | 77.8 |

**Table 5 Ablation studies of U-Net-like architecture in DSA performed on PoseTrack2017 validation set.**

| Method | mAP |
| --- | --- |
| r/m U-Net-like architecture | 75.9 |
| DSA, complete | 77.8 |

### Local information enhancement block

We explore the contribution of the proposed local information enhancement block. Removing this block results in a noticeable dip of 1.3 mAP, which provides empirical evidence that our proposed LEB is effective for extracting fine rearranged information of adjacent frames to improve the robustness of feature representation and preserve the geometric structure of local details.

### U-Net-like architecture

As shown in Table 5, we study the influence of the symmetric U-Net-like architecture. Specifically, we construct a baseline that performs cascaded global-to-local feature alignment within six layers. This baseline significantly reduces the performance by 1.9 mAP, with the final mAP 75.9. This demonstrates the effectiveness of the proposed symmetric U-Net-like architecture.

## DISCUSSION

The proposed dense symmetric temporal alignment (DSA) framework demonstrates significant advancements in video-based human pose estimation by effectively addressing the challenges of temporal misalignment and noisy feature aggregation in dynamic scenarios. The integration of global and local alignment mechanisms allows the model to refine temporal features hierarchically, outperforming existing methods on PoseTrack2017, PoseTrack2018, and a self-constructed badminton dataset. These results demonstrate that explicit temporal alignment mitigates noise from rapid motion and occlusion while preserving structural consistency. By decoupling global and local alignment, the method provides a modular framework that could integrate with other architectures to enhance long-range dependency modelling. Furthermore, the strong performance on the badminton dataset highlights its applicability in sports analytics,

where rapid motions and occlusions are prevalent. This suggests potential use cases in real-time athlete performance monitoring or rehabilitation systems. Future work could integrate lightweight detectors or explore end-to-end architectures to enhance efficiency. Additionally, we will extend temporal reasoning beyond adjacent frames, which may further improve robustness in complex multi-person interactions. Overall, DSA advances pose estimation and highlights the broader potential of symmetric temporal alignment in feature learning for achieving accurate pose estimation.

## CONCLUSION

In this article, we investigate the video-based human pose estimation task from the perspective of effectively leveraging support contexts from adjacent frames *via* symmetry feature alignment. We propose a dense symmetric temporal alignment U-Net-like framework to progressively rearrange the supporting frames with the current frame. Specifically, we design a global information alignment block to explore global-related connections that allow abundant auxiliary information to be aggregated from the supporting frames. Our local information enhancement block further aligns local details and produces meticulous fine-tuning at the pixel level to effectively deliver comprehensive representation. Extensive experiments confirm that our method significantly surpasses existing work on three different benchmark datasets, including PoseTrack2017, PoseTrack2018, and the self-constructed badminton dataset. In the future, we will extending the proposed DSA framework to other video-related tasks such as 3D human pose estimation and action recognition. The symmetric feature alignment approach can also be integrated into existing pose-tracking pipelines to enhance motion similarity assessment for improved data association. Additionally, exploring lightweight adaptations of the model for real-time applications on edge devices could further broaden its practical utility.

### Funding
The authors received no funding for this work.

### Competing Interests
Xingchen Wu is employed by Jiushao Institute of AI Algorithm, Jihua Laboratory.

### Author Contributions
- Guang Xv conceived and designed the experiments, performed the experiments, analyzed the data, performed the computation work, prepared figures and/or tables, authored or reviewed drafts of the article, and approved the final draft.
- Xingchen Wu conceived and designed the experiments, performed the experiments, analyzed the data, performed the computation work, prepared figures and/or tables, authored or reviewed drafts of the article, and approved the final draft.

## Data Availability

This code is available in the Supplemental Files.

The Badminton Dataset is available at GitHub and Zenodo:

- https://github.com/tzwx/Badminton.git.

- tzwx. (2025). tzwx/Badminton: Badminton Dataset (data). Zenodo. https://doi.org/10.5281/zenodo.15666349.

The PosTrack Dataset is available at Figshare:

https://doi.org/10.6084/m9.figshare.29876777.

## Supplemental Information

Supplemental information for this article can be found online at http://dx.doi.org/10.7717/peerj-cs.3100#supplemental-information.

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
