# Peer review of "Dense symmetric temporal alignment learning for human pose estimation"

_PeerJ Computer Science, doi:10.7717/peerj-cs.3100_

## Round 0.1 · original submission · Major Revisions

In the opinions of reviewers and mine, this paper should undertake a major revision to address those concerns.

**Language Note:** The review process has identified that the English language must be improved. PeerJ can provide language editing services - please contact us at [email protected] for pricing (be sure to provide your manuscript number and title). Alternatively, you should make your own arrangements to improve the language quality and provide details in your response letter. – PeerJ Staff

Reviewer 1 ·

Basic reporting

Thanks so much for your manuscript submission to Peer Journal. This paper proposed presents a model to Dense symmetric temporal alignment learning for human pose estimation. It is readable and containing some useful contributions to this journal. There are a few majors aspects of potential improvements which demands further updates (but not limited to) as I specified as below:
1. While the abstract has aimed to provide a comprehensive overview of the main contribution, there is a need to be revised so that the general reader can grasp the main idea/topic of the draft and the main contribution.
2. In the Introduction section, the main contribution and motivation is not clear; so, the authors should include a subsection to describes the contributions and motivations. It will be helpful for the reader to understand the novelty of their work.
3. The authors should include a new section, to discuss the some relevant and significant preliminaries which is used throughout the paper, and also include a notation table to define each notation properly, it would be helpful to understand the model.
4. In the related work section, a comparative analysis table can be included.

5. The authors tried to explain their results which is good but what I suggest the authors to follow this method: first show the results in form of figure or table, explain the behavior’s of the results, explaining why and how this behavior’s happen and lastly which most importantly to justify their result by comparing with existing journal articles.

6. The authors, please correct grammatical errors and also maintain orientation throughout the paper. The manuscript has so much grammatical and orientation relate issues, it shows the authors have not proper read it before submission. The authors please properly arranged the manuscript in the correct way.
7. What is next? What are future directions? Authors should include some future work of their model.

Experimental design

NA

Validity of the findings

NA

Cite this review as

Reviewer 2 ·

Basic reporting

N/A

Experimental design

N/A

Validity of the findings

N/A

Additional comments

This article presents a novel framework, Dense Symmetric Temporal Alignment (DSA), for video-based human pose estimation, aiming to address challenges such as rapid motion and occlusion by hierarchically aligning global and local temporal features across frames. While the work offers promising advancements in temporal feature aggregation, the comments below may help the authors improve the article’s quality.
1. The abstract and introduction redundantly repeat the same content, which diminishes conciseness and readability.
2. The literature review lacks discussion of recent works (post-2022) on temporal alignment in pose estimation, weakening contextualization of the method’s novelty.
3. The methodological distinction between the proposed temporal deformable convolution in GAB and prior uses of deformable convolutions for alignment is insufficiently clarified.
4. The article “RA-UNet: An improved network model for image denoising” could strengthen the methodological discussion of your U-Net-like architecture by providing comparative insights into how symmetric encoder-decoder designs enhance feature aggregation and noise suppression in tasks requiring fine-grained spatial alignment, such as pose estimation in occluded scenarios.
5. The self-built badminton dataset lacks critical details on annotation protocols, inter-rater reliability, and diversity of motion scenarios, raising concerns about bias and generalizability.
6. The rationale for selecting specific hyperparameters (e.g., 25% bounding box enlargement, 4 supporting frames) is not empirically or theoretically justified.
7. Comparisons with recent state-of-the-art methods (e.g., 2023 works) are absent, weakening claims of superiority.
8. The ablation study does not evaluate the impact of the symmetric U-Net-like architecture itself, leaving its contribution to performance unclear.
9. Citing the article "CGTracker; A center graph based multi-pedestrian-object detection and tracking" would contextualize your model’s performance on multi-person pose estimation in dynamic environments (e.g., the badminton dataset) by referencing advanced graph-based methods for resolving spatial ambiguities and occlusions in crowded scenes.
10. The justification for using L2 loss over alternatives (e.g., focal loss) for heatmap regression is not provided, despite its relevance to class imbalance.
11. The discussion section does not address limitations such as performance on heavily occluded joints or computational overhead compared to baseline methods.
12. The conclusion reiterates results without synthesizing broader implications for real-world applications or future research directions.
13. The novelty of symmetric alignment is not rigorously contrasted with prior symmetric architectures (e.g., U-Net variants in pose estimation), weakening claims of originality.
14. The reference "Information Propagation Prediction Based on Spatial–Temporal Attention and Heterogeneous Graph Convolutional Networks" aligns with your temporal alignment framework’s focus on hierarchical spatio-temporal dependencies, offering theoretical support for integrating attention mechanisms to model inter-frame dynamics in video-based pose estimation.
15. The local enhancement block’s transformation matrix Q and vector k are described without addressing potential overfitting risks in adaptive convolution.

Cite this review as

·

Basic reporting

Lines 57-62: Run-on sentence with unclear meaning: "Existing video-based algorithms directly aggregate information from neighboring frames to increase the information abundance of the target frame.
Line 60: Should be "rapid movement" not "rapidly movement"
Lines 82-83: Awkward phrasing: "Our goal is to better leverage neighboring frame sequences through a principled global and local feature alignment.
Inconsistent notation: The paper uses both "It-1" and "Ft-1" notation styles throughout, sometimes with subscripts and sometimes with regular text indices.
The paper cites relevant works but does not critically analyze their limitations in sufficient depth, you can include more relevant and latest references for example: 10.1016/j.heliyon.2024.e32628 10.1049/ell2.12806 10.3390/app13042462 10.3390/s24041273
The paper claims to solve the problem of "directly incorporating unaligned additional visual cues from adjacent frames" (lines 13-15), but doesn't clearly quantify how much improvement is specifically due to better alignment versus other factors.
The connection between the method components (GAB and LEB) and the specific challenges they address could be more explicitly stated and evaluated.

Experimental design

Figure 1: The diagram is overly crowded with small text that is difficult to read. The legend at the bottom contains too many symbols crammed together.
Tables 1, 2, and 3: Alignment issues are present, particularly with numerical entries. The tables should use consistent decimal precision across all values.
Figure 2 and 3: The visual results lack annotations pointing out specific improvements or challenging areas. Comparison examples showing the difference between the proposed method and others would significantly strengthen these figures.

Validity of the findings

Lines 116-122: The definition of parameters θ and M is incomplete, and the notation is inconsistent.
Line 122: The regular convolution kernel ℜ is represented with a non-standard notation and its definition is unclear.
Equation 3 (line 128): The notation is inconsistent with other equations and some terms are not properly defined.
Line 146-147: The transformation matrix Q and translation vector k are introduced without sufficient explanation of their dimensions or constraints.
Equation 6 (line 147): The notation for adaptive convolution is unclear, particularly the relationship between offset parameters O and the computation of F̄t-1.

Additional comments

The paper presents a novel approach for video-based human pose estimation using a symmetric U-Net-like architecture with global and local alignment mechanisms to effectively utilize temporal information from adjacent frames, will consider after the major revision.

---

## Round 0.2 · accepted · Accept

In the opinions of original reviewers and mine, this revised paper is acceptable for publication.

Reviewer 1 ·

Basic reporting

The authors have addressed all my comments, no further comments. So, I vote for acceptance.

Experimental design

NA

Validity of the findings

NA

Additional comments

NA

Cite this review as

Reviewer 2 ·

Basic reporting

N/A

Experimental design

N/A

Validity of the findings

N/A

Additional comments

N/A

Cite this review as

·

Basic reporting

The manuscript now adheres well to academic writing standards. The revised abstract is clearer and more concise, and the overall structure follows the discipline’s norms. Figures and tables are relevant and appropriately referenced.

Experimental design

The research question is well-defined and directly addresses shortcomings in temporal alignment in video-based pose estimation. The authors propose an effective architectural solution, and the methodology is sound.

Validity of the findings

The results are robust and validated across three datasets. The ablation studies demonstrate the importance of key model components.This work makes a meaningful contribution to pose estimation in videos, particularly under occlusion and motion blur. It is ready for publication in its current form.

Additional comments

This work makes a meaningful contribution to pose estimation in videos, particularly under occlusion and motion blur. It is ready for publication in its current form.